# Event-3DGS: Event-based 3D Reconstruction Using 3D Gaussian Splatting

**Hanqian Han**   **Jianing Li**   **Henglu Wei**   **Xiangyang Ji**[*]
Tsinghua University

## Abstract

Event cameras, offering high temporal resolution and high dynamic range, have brought a new perspective to addressing 3D reconstruction challenges in fast-motion and low-light scenarios. Most methods use the Neural Radiance Field (NeRF) for event-based photorealistic 3D reconstruction. However, these NeRF methods suffer from time-consuming training and inference, as well as limited scene-editing capabilities of implicit representations. To address these problems, we propose Event-3DGS, the first event-based reconstruction using 3D Gaussian splatting (3DGS) for synthesizing novel views freely from event streams. Technically, we first propose an event-based 3DGS framework that directly processes event data and reconstructs 3D scenes by simultaneously optimizing scenario and sensor parameters. Then, we present a high-pass filter-based photovoltage estimation module, which effectively reduces noise in event data to improve the robustness of our method in real-world scenarios. Finally, we design an event-based 3D reconstruction loss to optimize the parameters of our method for better reconstruction quality. The results show that our method outperforms state-of-the-art methods in terms of reconstruction quality on both simulated and real-world datasets. We also verify that our method can perform robust 3D reconstruction even in real-world scenarios with extreme noise, fast motion, and low-light conditions. Our code is available in `https://github.com/lanpokn/Event-3DGS`.

## 1 Introduction

3D reconstruction [8] plays a crucial role in various cutting-edge fields, such as robot vision, virtual reality, and augmented reality systems. It usually enables the creation of accurate 3D models from ideal frame sequences. Nevertheless, with conventional cameras, 3D reconstruction performance has suffered from a significant drop in some challenging conditions [13, 36] (e.g., fast motion blur and low light). Thus, how to use a new visual sensing paradigm for 3D reconstruction to overcome the shortcomings of conventional cameras remains a partially unsolved issue.

Event cameras [9, 21, 30], namely bio-inspired dynamic vision sensors, fundamentally differ from conventional cameras that capture frames at fixed intervals. Event cameras operate asynchronously, recording light changes with dynamic events at the microsecond level. This unique property endows event cameras with high temporal resolution, high dynamic range, low power consumption, and low latency. These advantages have driven their application in various challenging vision tasks [12, 20, 23, 41, 43, 50], including recent efforts in 3D reconstruction [32].

Despite efforts [3, 17, 42, 51] to use event cameras for 3D reconstruction, real-world performance in terms of quality, robustness, and real-time capabilities still needs improvement. Traditional non-learning optimization-based methods [3, 17, 32, 51] serve as the foundation for event-based 3D reconstruction, but they often struggle with robustness and rendering quality. Recently, Neural

---

[*]Corresponding author: xyji@tsinghua.edu.cn

38th Conference on Neural Information Processing Systems (NeurIPS 2024).

Radiance Fields (NeRF) [10, 28, 40] have gained popularity for scene representation and novel view synthesis from event data, utilizing a Multi-Layer Perceptron (MLP) and differentiable rendering. Although these NeRF-based methods [1, 2, 6, 14, 18, 24, 25, 26, 27, 31, 35, 52] achieve impressive results in photorealistic 3D reconstruction from neuromorphic cameras, they suffer from time-consuming training and inference processes. Additionally, their implicit representations limit scene editing capabilities. Moreover, NeRF have primarily been investigated using simulated data and high-quality real-world images captured under ideal conditions (e.g., optimal lighting and minimal noise), posing limitations on real-world 3D reconstruction. In contrast, the emergence of 3D Gaussian Splatting (3DGS) [5, 11, 15, 46, 47] presents a compelling alternative, boasting high reconstruction accuracy and swift inference speed. However, 3DGS has predominantly been utilized with image or video data for 3D reconstruction, with limited exploration in event streams.

To address this gap, we propose Event-3DGS, the first event-based reconstruction framework utilizing 3DGS for synthesizing novel views from event streams. More specifically, we introduce an event-based 3DGS framework, enabling direct processing of event data and reconstruction of 3D scenes while simultaneously optimizing scenario and sensor parameters. Then, we present a high-pass filter-based photovoltage estimation module, effectively reducing noise in event data to enhance the robustness of our method in real-world scenarios. Finally, we propose an event-based 3D reconstruction loss to optimize the parameters of our method for better reconstruction quality. Extensive experiments show that our method outperforms state-of-the-art methods in reconstruction quality on simulated and real-world datasets. This pioneering work in event-based 3D reconstruction with 3DGS sets a new benchmark, opening new avenues for high-quality, efficient, and robust 3D reconstruction in challenging real-world scenarios, such as extreme noise, fast motion, and low light. Our contributions can be summarized as follows:

- We introduce Event-3DGS, the first framework that combines event cameras with 3DGS technology, enabling 3D reconstruction in challenging real-world scenarios.

- We present a high-pass filter-based photovoltage contrast estimation module, which effectively estimates photovoltage contrast by reducing noise in event streams for robust 3D reconstruction.

- We design a novel event-based 3D reconstruction loss to optimize the parameters of our method for better reconstruction quality.

## 2    Related Works

**Event-based 3D Reconstruction.** Early attempts [3, 16, 17, 32, 51] at using event cameras for 3D reconstruction typically relied on geometric models and handcrafted features. However, these non-learning, optimization-based methods often struggle to achieve robust and high reconstruction quality. A growing trend is the use of neural radiance fields (NeRF) for scene representation and novel view synthesis. For instance, Ev-NeRF [14] and E-NeRF [18] are some of the earliest works to apply NeRF for event-based 3D reconstruction. These methods render images at different times, generate events through differencing, and compare them with actual events. Further advancements include DeNeRF [25] and EvDNeRF[1], which introduce Deformable NeRF for dynamic scene reconstruction. EventNeRF [35] extends this by enabling colored rendering through the incorporation of three-channel events into NeRF. Some methods like E2NeRF [31] and Ev-DeblurNeRF [2] perform hybrid reconstruction to mitigate motion blur by combining blurred images with events. However, these NeRF-based approaches face significant challenges, including the time-consuming generation of novel views and limited scene editing capabilities due to implicit representations. Additionally, NeRF have primarily been explored using simulated data and high-quality images captured under ideal conditions, leaving a considerable gap between the models and real-world scenarios. Therefore, our goal is to design a novel event-based 3D reconstruction framework that ensures high-quality, efficient, and robust performance in real-world scenarios.

**3D Gaussian Splatting for 3D Reconstruction.** 3D Gaussian Splatting (3DGS) [15] has significantly advanced in 3D scene representation, offering notable advantages over NeRF by capturing complex geometries and lighting effects more accurately and efficiently. These advantages make 3DGS highly suitable for real-time and real-world applications. Extended works like 4DGS [46, 47] and D-3DGS [48] further enhance dynamic scene rendering. Besides, some works integrate with Simultaneous Localization and Mapping (SLAM) [39] and text-to-3D models [4] to expand 3DGS capabilities. However, these methods have primarily been applied to image and video data, leaving

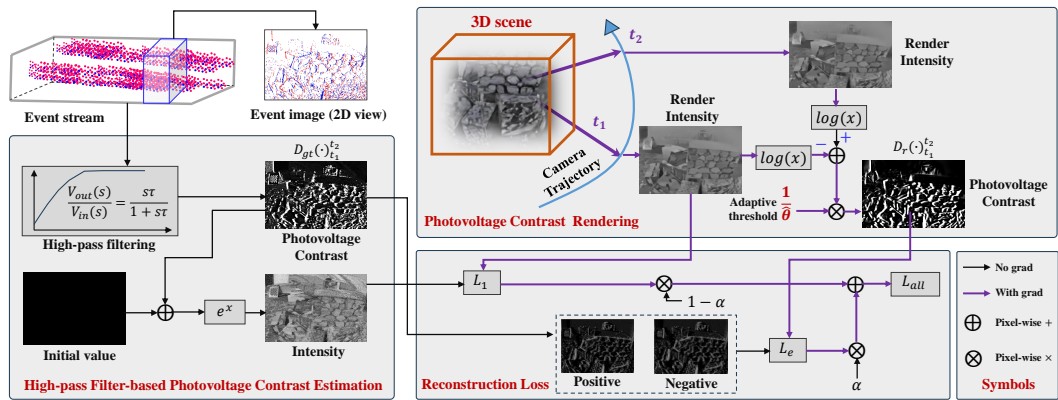

Figure 1: The pipeline of **Event-based 3D Reconstruction using 3D Gaussian Splatting (Event-3DGS)**. The proposed event-based 3DGS framework enables direct processing of event data and reconstructs 3D scenes while simultaneously optimizing scenario and sensor parameters. A high-pass filter-based photovoltage contrast estimation module is presented to reduce noise in event data, enhancing the robustness of our method in real-world scenes. An event-based 3D reconstruction loss is designed to optimize the parameters of our method for better reconstruction quality.

their potential with event cameras largely unexplored. Thus, designing a novel 3DGS model to directly process asynchronous events for 3D reconstruction remains an open challenge.

## 3 Method

### 3.1 Event-3DGS Architecture

To achieve high-quality, efficient, and robust 3D reconstruction in challenging real-world scenarios, we propose a novel event-based 3D reconstruction framework using 3D Gaussian Splatting (Event-3DGS). As shown in Fig. 1, our framework mainly consists of three modules: **high-pass filter-based photovoltage contrast estimation**, **photovoltage contrast rendering**, and **event-based 3D reconstruction loss**. More precisely, we first present a high-pass filter-based photovoltage contrast estimation module that reduces noise in event data to enhance the robustness of our method in real-world scenes (see Sec. 3.2). Then, we design a photovoltage contrast rendering module that obtains the photovoltage contrast image by calculating the difference in light intensity in 3DGS. After obtaining two contrast estimations, we propose a novel event-based 3D reconstruction loss to measure the differences (see Sec.3.3). Finally, our method optimizes the 3D scene and camera parameters by propagating gradients through backpropagation.

3DGS [15] demonstrates superior 3D reconstruction capabilities by rapidly converting input images into highly detailed 3D point clouds, accurately representing the scene. For a specific 3D scene represented by 3D Gaussian functions, the forward process of 3DGS can be regarded as a mapping function $G(\mathbf{T})$, which gets the rendered image by alpha blending in the corresponding camera pose $\mathbf{T}$ in time $t$. For a single pixel on a single channel, alpha blending can be described as follows:

$$L = \sum_{i=1}^{N} l_i \alpha_i \prod_{j=1}^{i-1}(1 - \alpha_j),\tag{1}$$

where $L$ denotes the pixel value result, which can be intensity or one of the three channels. $l_i$ and $\alpha_i$ are the color and opacity of each point mapped to this pixel, respectively.

Event cameras operate on a fundamentally distinct imaging principle, generating event data in the form of sparse points (see Sec. 3.2). This disparity prevents the direct integration of asynchronous events into the original 3DGS formulation. To bridge this gap, we integrate event data seamlessly with the output of 3DGS by leveraging photovoltage contrast (i.e., intensity changes). Considering two close-in-time instances $t_1$ and $t_2$, the camera poses corresponding to these two moments are $\mathbf{T_1}$ and $\mathbf{T_2}$. We can obtain its photovoltage contrast between two moments using the proposed high-pass

filter-based photovoltage contrast estimation module, and it can be formulated as:

$$D_{gt}(\cdot)_{t_1}^{t_2} = \frac{1}{\theta}((V(\cdot, t_2) - V(\cdot, t_1)), \tag{2}$$

where $V(p, t)$ refers to the photovoltage in pixel $p$ and time $t$.

Correspondingly, we need to use 3DGS to render the photovoltage contrast and light intensity, and then compare these with the ground truth obtained from the event data for subsequent reconstruction. The photovoltage contrast image can be obtained by the proposed photovoltage contrast rendering module. The rendering process can be mathematically described as follows:

$$D_r(\cdot)_{t_1}^{t_2} = \frac{1}{\hat{\theta}}(\log(G(\mathbf{T_2}) + \epsilon) - \log(G(\mathbf{T_1}) + \epsilon)), \tag{3}$$

where $G(\mathbf{T})$ denotes the intensity result in 3DGS. $D_r(p)$ is the normalized contrast value in the pixel p, and $\epsilon$ is a small number that avoids log(0). $\hat{\theta}$ is a learnable parameter, which is an estimate of the threshold in the event sensor. For the sake of convenience in writing, we do not strictly distinguish between time $t$ and the corresponding pose $T$.

Given the intrinsic characteristics of event data, it's essential to highlight that intensity estimation methods [37, 45] are unlikely to outperform photovoltage contrast estimation techniques. Even data-driven learning-based models [7, 34] may only yield visually appealing results without ensuring physical accuracy and generality. To address this challenge, we proposed a dynamic adjustment strategy for intensity to use high-quality photovoltage while ensuring robustness and stability.

### 3.2 High-pass Filter-based Photovoltage Contrast Estimation

To gain a deeper understanding of the high-pass filter-based photovoltage contrast estimation module, it's essential to begin with a fundamental understanding of the Dynamic Vision Sensor (DVS) [9]. The DVS is designed to capture changes in light intensity pixel-wise and asynchronously. It accomplishes this by converting the light spectrum into photocurrent as follows:

$$I(p, t) = \int \lambda \cdot QE(\lambda) \cdot L(p, t, \lambda) \cdot d\lambda, \tag{4}$$

where $L(p, t, \lambda)$ denotes the spectrum, $t$ is time, $p$ refers to the pixel coordinate $\{x, y\}$, and $\lambda$ is the wavelength of light. $I(p, t)$ is the photocurrent or intensity in pixel $p$ and time $t$. $QE$ is quantum efficiency, which represents the weighting of different wavelength light converted into photocurrent.

Subsequently, the photocurrent will be converted into photovoltage through a logarithmic function. When the voltage change surpasses a predefined threshold $\theta$, an event will be triggered [30] as:

$$V(p, t_2) - V(p, t_1) = \sigma_i^p \theta, \tag{5}$$

where $V(p, t)$ refers to the logarithmic operation of the photocurrent $I(p, t)$. $\sigma_i^p$ is either +1 or -1 by comparing $V(p, t_2)$ and $V(p, t_1)$.

In general, the threshold $\theta$ is an inherent attribute of the DVS. It can be defined as follows:

$$\theta = \frac{C_{diff}V_{th}}{U_T A_v}, \tag{6}$$

where $C_{diff}$ is determined by the capacitance in the ,$V_{th}$ is a fixed constant, $U_T$ is thermal voltage, and $A_v$ is voltage gain factor. This formula illustrates that $\theta$ often varies with sensor settings and environmental changes, making it generally difficult to obtain its true value. In this work, we will optimize the threshold $\theta$ together with the 3D scene.

Intuitively, asynchronous events appear as sparse points [19] in the spatiotemporal domain:

$$E(p, t) = \sum_{i=1}^{N} \sigma_i^p \theta \delta(t - t_i^p), \tag{7}$$

where $\delta(\cdot)$ refers to the Dirac delta function, with $\int \delta(t)\, dt = 1$ and $\delta(t) = 0, \forall t \neq 0$.

In this work, we implement the reverse process of dynamic event generation by extracting photovoltage contrast and intensity from the event stream $E(p, t)$. The photovoltage is given as:

$$V(p, t) - V(p, t_i) = E(p, t_j) + w(p, t), \tag{8}$$

where $t_i$ and $t_j$ are the times corresponding to two adjacent events around $t$. $w(p, t)$ denotes an uncertainty term, which is determined by the mathematical principles of the event camera and is unrelated to noise. This formula indicates that the event camera cannot accurately reconstruct the intensity of light outside the triggering event's timing. Moreover, since $V(p, t_i)$ is unknown, it is also impossible to accurately obtain the intensity of light at each event's timing. In short, only the intensity difference between each event's triggering times can be accurately obtained.

For the photovoltage between two events, we can simply assume:

$$w(p, t) = -E(p, t_j), t < t_j. \tag{9}$$

The above assumption will not interfere with the voltage at the time of event triggering. If $E(p, t)$ is ideal, V can be directly obtained from $E$ through pure integration, that is:

$$\hat{V}_d(p, t) = V(p, t) - V(p, t_0) = \int_0^t E(p, t)dt, \tag{10}$$

where $\hat{V}_d(p, t)$ is the photocurrent contrast to be estimate. Ideally, this method can accurately provide the photovoltage contrast between the triggering moments of each event.

However, when event cameras are applied in real-world 3D reconstruction, they often encounter various types of noise, making it challenging to accurately estimate the photovoltage contrast through pure integration [37]. To address this issue, we use high-pass filtering to process the event data. The high-pass filtering can be depicted as follows:

$$\frac{\mathcal{V}_{out}(s)}{\mathcal{V}_{in}(s)} = \frac{s\tau}{1 + s\tau}, \tag{11}$$

where $V_{out}(s)$ and $V_{in}(s)$ are the input-output signals after the Laplace transformation. $\tau$ is the time constant related to the cutoff frequency. We treat $\int_0^t E(p, t)dt$ as a noisy input $V_{in}$ and $\hat{V}_d(p, t)$ as the output $V_{out}$. When substituted into Eq. 11 and transformed back into the time domain, we obtain that:

$$\hat{V}_d(p, t) = E(p, t) - \frac{1}{\tau}\dot{\hat{V}}_d(p, t), \tag{12}$$

where $\dot{\hat{V}}_d(p, t)$ is the differential with respect to the timestamp.

By simultaneously solving with Eq. 7, we can estimate the photovoltage contrast between any two corresponding moments. Once the photovoltage is obtained, the intensity can be computed as:

$$I(p, t) = e^{\hat{V}_d(p,t) + \hat{V}_d(p,0)}. \tag{13}$$

$V(p, t_0)$ does not affect photovoltage contrast estimation. Therefore, our high-pass filter-based method typically provides more accurate results than restoring pure photovoltage or intensity.

### 3.3 Event-based 3D Reconstruction Loss

For better reconstruction quality, we use a loss function with two key components: intensity estimation and photovoltage estimation. The light intensity is evaluated as follows:

$$l_i^{t_1} = l_1(I(\cdot, t_1), G(\mathbf{T_1})), \tag{14}$$

where $l_1$ is a loss function that computes the average of the absolute values between each pixel.

For the evaluation of photovoltage contrast, it's important to consider that 3DGS uses a rasterization method to generate the entire image at once. This means the rendering sampling time often does not match the event triggering time for most pixels. According to Eq. 8, the ground truth of the photovoltage contrast inherently contains some errors. Additionally, despite applying filtering methods, event data still has significant noise that cannot be entirely eliminated. Consequently, if we directly use the $L_1$ loss to compare the rendered photovoltage contrast with the photovoltage contrast calculated from event data, these errors will be strictly considered during the reconstruction process,

Table 1: Performance comparison on the DeepVoxels synthetic dataset [38]. our Event-3DGS outperforms two state-of-the-art methods and our baseline using pure integration without filtering.

| Sequence | E2VID [34] | | | E2VID [34]+3DGS [15] | | | PI-3DGS | | | Event-3DGS | | |
|---|---|---|---|---|---|---|---|---|---|---|---|---|
| | SSIM | PSNR | LPIPS | SSIM | PSNR | LPIPS | SSIM | PSNR | LPIPS | SSIM | PSNR | LPIPS |
| mic | 0.938 | 19.965 | 0.048 | 0.946 | 19.955 | 0.068 | 0.955 | 21.979 | 0.060 | 0.952 | 21.127 | 0.063 |
| ship | 0.808 | 16.556 | 0.108 | 0.825 | 16.681 | 0.122 | 0.792 | 16.750 | 0.177 | 0.818 | 17.815 | 0.147 |
| materials | 0.872 | 18.302 | 0.084 | 0.885 | 18.325 | 0.094 | 0.925 | 20.053 | 0.062 | 0.933 | 20.506 | 0.060 |
| lego | 0.883 | 19.744 | 0.075 | 0.899 | 20.002 | 0.084 | 0.928 | 23.853 | 0.056 | 0.925 | 23.046 | 0.058 |
| ficus | 0.932 | 19.795 | 0.043 | 0.935 | 19.626 | 0.056 | 0.939 | 19.880 | 0.050 | 0.940 | 19.939 | 0.049 |
| drums | 0.908 | 18.312 | 0.071 | 0.915 | 18.288 | 0.085 | 0.953 | 22.643 | 0.041 | 0.951 | 22.568 | 0.042 |
| chair | 0.939 | 23.842 | 0.040 | 0.949 | 23.866 | 0.050 | 0.954 | 27.024 | 0.042 | 0.953 | 27.336 | 0.050 |
| Average | 0.897 | 19.502 | 0.067 | 0.908 | 19.535 | 0.080 | 0.921 | 21.740 | 0.070 | **0.925** | **21.762** | **0.067** |

which can actually degrade the reconstruction quality. Thus, if a new loss can be designed that allows for a certain tolerance in the estimation of photovoltage contrast, it would enable better completion of the reconstruction task. For simplification, we denote $L_g^p = D_{gt}(p)_{t_1}^{t_2}$, $L_r^p = D_r(p)_{t_1}^{t_2}$. The loss function of photovoltage contrast can be formulated as:

$$l_e^{p,t_1,t_2}(L_g^p, L_r^p) = \begin{cases} R(|L_r^p - L_g^p - \beta| - \beta) & \text{if } L_g^p > 0 \\ R(|L_g^p - L_r^p - \beta| - \beta) & \text{if } L_g^p < 0 \end{cases}, \tag{15}$$

where $R(x) = \max(0, x)$, and $\beta$ is a measure of the tolerance in the estimated photovoltage contrast obtained from rendering. When $\beta$ is infinitely large, the loss becomes a trivial constant zero, imposing no constraints. Conversely, if $\beta$ is zero, the loss degenerates into a very strict $L_1$ loss. Theoretical analysis shows that setting $\beta$ to 0.5 yields excellent results in the real-world dataset. When $\beta$ is set to 0.5, it best aligns with the characteristics described in Eq. 8.

Finally, the total loss function for event-based 3D reconstruction can be described as follows:

$$l_{all}(t_1, t_2) = \alpha \sum_{p \in R^2} \frac{l_e^{p,t_1,t_2}(L_g^p, L_r^p)}{W * H} + (1 - \alpha)l_i^{t_1}, \tag{16}$$

where W and H are the width and height of the image, respectively. $\alpha$ is a parameter that controls the weight of the intensity. For the first 8000 epoch, $\alpha$ is set to 0 to give an initialization of the reconstruction. Then, $\alpha$ is generally set near 1 to improve the quality of 3D reconstruction.

## 4 Experiments

### 4.1 Experimental Setting

**Datasets.** To evaluate the effectiveness of our Event-3DGS, we conduct experiments on the DeepVoxels synthetic dataset [38] and the real-world Event-Camera dataset[29]. For the synthetic dataset, we use seven sequences with continuous 180-degree image rotations on a gray background as the ground truth for reconstruction. These sequences are processed by the VOLT simulator [22] to generate event data, offering a more realistic simulation than ESIM [33] with higher noise levels. For the real-world dataset, we select five typical sequences that provide aligned image and event data under fast motion and low-light conditions. For longer sequences, we typically utilize the initial 100 images for training and evaluate performance on separate data not employed during reconstruction.

**Implementation Details.** We set $\tau$ to 0.05 for the high-pass filter-based photovoltage contrast estimation module. In the loss function, we set $\alpha$ to 0.9. For synthetic experiments with low noise, $\beta$ is set to 0, while for real data with higher noise, $\beta$ is set to 0.5. We utilize E2VID [34] for initial intensity estimation. We use E2VID+3DGS as a baseline for event-based 3D reconstruction, comparing it with our full method to validate its efficacy. All experiments are conducted on an AMD Ryzen Threadripper 3970X 32-Core CPU and an NVIDIA GeForce RTX 3080 Ti GPU. The evaluation metrics use the Peak Signal Noise Ratio (PSNR), the Structural Similarity (SSIM) [44], and the Learned Perceptual Image Patch Similarity (LPIPS) [49].

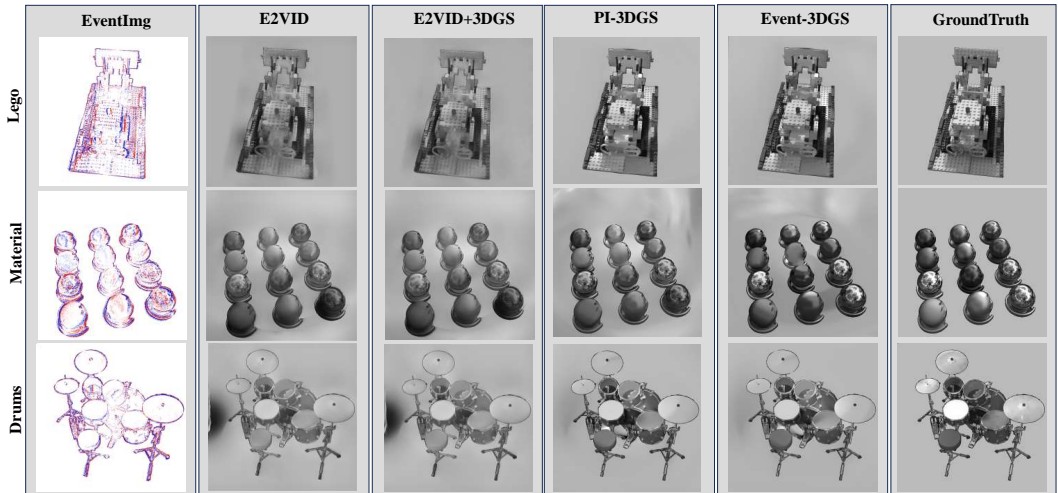

Figure 2: Representative visualization results on the DeepVoxels synthetic dataset [38]. Obviously, our Event-3DGS produces visually pleasing images with fine details and fewer artifacts.

Table 2: Performance comparison on the real-world Event-Camera dataset[29]. Note that, our Event-3DGS surpasses three comparative methods on three metrics.

| Sequence | E2VID [34] | | | E2VID [34]+3DGS [15] | | | PI-3DGS | | | Event-3DGS | | |
|---|---|---|---|---|---|---|---|---|---|---|---|---|
| | SSIM | PSNR | LPIPS | SSIM | PSNR | LPIPS | SSIM | PSNR | LPIPS | SSIM | PSNR | LPIPS |
| boxes | 0.356 | 8.705 | 0.320 | 0.408 | 8.841 | 0.228 | 0.224 | 8.364 | 0.749 | 0.575 | 17.696 | 0.260 |
| office_zigzag | 0.326 | 7.550 | 0.287 | 0.346 | 7.649 | 0.233 | 0.310 | 10.009 | 0.424 | 0.430 | 14.043 | 0.183 |
| slider_depth | 0.353 | 7.634 | 0.317 | 0.356 | 7.620 | 0.309 | 0.477 | 13.509 | 0.276 | 0.497 | 12.448 | 0.261 |
| outdoors_walking | 0.156 | 3.657 | 0.508 | 0.179 | 3.499 | 0.361 | 0.138 | 3.429 | 0.738 | 0.271 | 10.583 | 0.300 |
| calibration | 0.239 | 5.669 | 0.293 | 0.276 | 5.742 | 0.316 | 0.270 | 11.698 | 0.714 | 0.312 | 11.065 | 0.222 |
| Average | 0.286 | 6.643 | 0.345 | 0.313 | 6.670 | 0.290 | 0.284 | 9.402 | 0.580 | **0.417** | **13.167** | **0.245** |

## 4.2 Effective Test

**Evaluation on Synthetic Data.** To verify the effectiveness of our method, we select two state-of-the-art methods (i.e., E2VID [34] and reconstructed images for 3DGS [15]) and our baseline using Pure Integration without filtering for 3DGS (PI-3DGS) as three comparison methods. As shown in Table 1, our method and our baseline outperform the representative reconstruction method (i.e., E2VID [34]) on the DeepVoxels synthetic dataset [38]. More precisely, our method has improved by 0.017 and 2.227 respectively compared to the competitor (i.e., E2VID [34]+3DGS [15]) in SSIM and PSNR, while decreasing by 0.013 in LPIPS. Furthermore, we present some representative visualization results on the DeepVoxels synthetic dataset [38] in Fig. 2. Note that, our method produces visually pleasing images with fine details and fewer artifacts. For clarification, the visualization is for comparative analysis, our method is capable of synthesizing novel views.

**Evaluation on Real-world Data.** To evaluate the performance of our method in real-world scenarios, we present the comparative results of the real-world Event-Camera dataset[29] in Table 2. We select three representative comparison methods to highlight the performance of Event-3DGS. Our baseline (i.e., PI-3DGS) performs worse because the pure integration method for estimating light intensity is sensitive to noise. Our method, with its high-pass filter-based photovoltage contrast estimation module, effectively filters out noise in real scenes and improves reconstruction quality (see Fig. 3). To demonstrate its robustness in extreme scenarios such as high-speed motion or low lighting, we selected two typical scenarios in Fig. 4. We can find that conventional cameras struggle in low-light conditions and produce significant motion blur in high-speed scenarios. These examples show that our method can reconstruct 3D scenes from the event stream to overcome the limitations of conventional cameras in challenging conditions.

## 4.3 Ablation Test

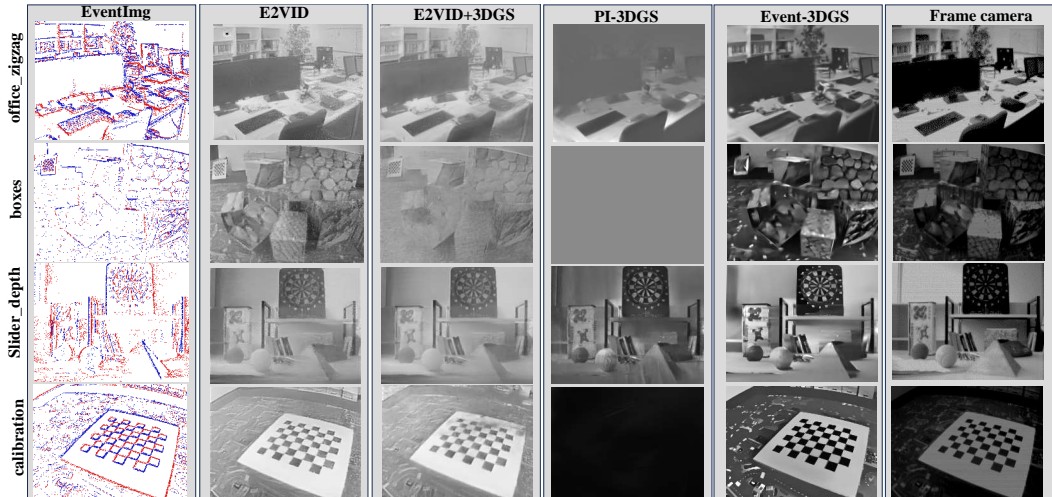

Figure 3: Representative visualization results on the real-world Event-Camera dataset[29]. Note that, our Event-3DGS achieves better reconstruction quality than the three comparative methods.

**Contribution of Each Component.** To explore the impact of each component on the final reconstruction performance, we chose the pure integration image without the adaptive threshold and event loss as the baseline. As illustrated in Table 3, We use three different strategies (e.g., adaptive thresholding, high-pass filtering, and loss function) to enhance our baseline. As a result, our method achieves the best performance among all competitors. In other words, our method employs these effective components to process event streams for 3D reconstruction.

**Influence of the Parameter $\alpha$.** To analyze the hyperparameter $\alpha$ of the loss function, we set the hyperparameter $\alpha$ with various values (e.g., 0.05, 0.2, 0.4, 0.6, 0.8, 0.9, and 1). As shown in Table 4, large values of $\alpha$ pose a risk of the training deviating to suboptimal points. Small values render the main phase ineffective, resulting in degraded outcomes similar to the initial phase.

**Influence of the Parameter $\beta$.** We report the impact of varying $\alpha$ in Table 5. The results indicate that the reconstruction performs optimally when $\beta$ is set to 0.5. Deviations from this value led to a decline in performance, consistent with theoretical predictions.

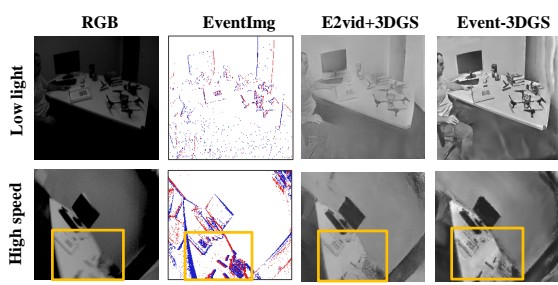

Figure 4: Representative visualization examples on low-light and high-speed motion blur scenarios.

Table 3: The contribution of each component.

| | | | | | | | | |
|---|---|---|---|---|---|---|---|---|
| Threshold | | | ✓ | | ✓ | ✓ | ✓ | |
| Filtering | | | ✓ | | ✓ | | ✓ | ✓ |
| Loss | | ✓ | | | ✓ | ✓ | | ✓ |
| SSIM | 0.219 | 0.317 | 0.376 | 0.221 | 0.410 | 0.310 | 0.393 | **0.430** |
| PSNR | 6.713 | 11.197 | 9.594 | 6.878 | 11.715 | 10.009 | 9.840 | **14.043** |
| LPIPS | 0.767 | 0.463 | 0.249 | 0.765 | 0.217 | 0.424 | 0.191 | **0.183** |

Table 4: The influence of the Parameter $\alpha$

| $\alpha$ | 1 | 0.99 | 0.9 | 0.8 | 0.6 | 0.4 | 0.2 | 0.05 |
|---|---|---|---|---|---|---|---|---|
| SSIM | 0.945 | 0.953 | **0.956** | 0.955 | 0.954 | 0.953 | 0.952 | 0.952 |
| PSNR | 25.688 | **27.302** | 25.974 | 25.066 | 24.540 | 24.300 | 24.146 | 24.148 |
| LPIPS | 0.063 | 0.051 | 0.039 | 0.040 | **0.038** | 0.038 | 0.039 | 0.039 |

## 4.4 Scalability Test

**Event-3DGS for Motion Deblurring.** Our Event-3DGS can be further expanded to motion deblurring. By integrating event data with RGB frames, our method can achieve deblurring effects using the hybrid reconstruction manner. As shown in Fig. 6, We test the hybrid framework on some simulated sequences [15] using VOLT [22]. Note that, a blurred image is generated through integration to serve as the RGB input. Our Event-3DGS leverages event data to achieve deblurred color reconstruction.

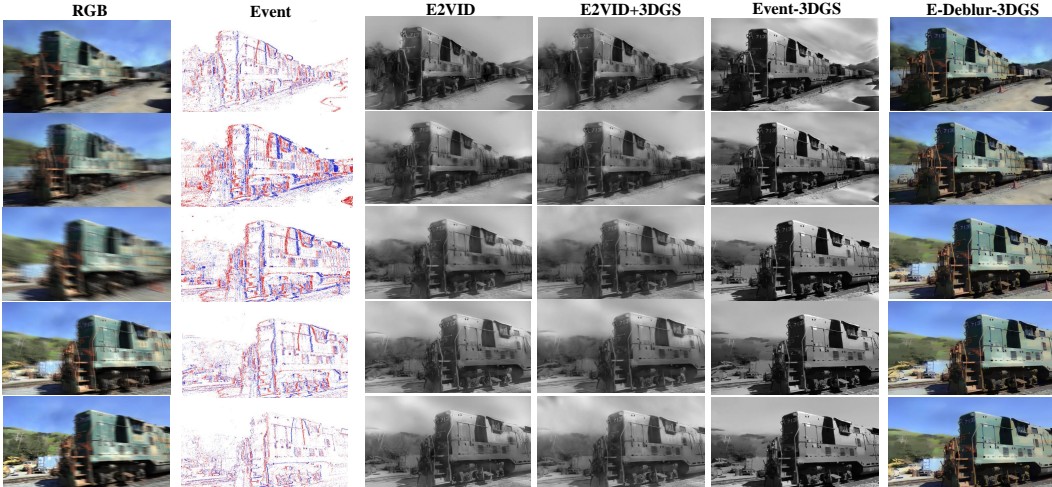

| RGB | Event | E2VID | E2VID+3DGS | Event-3DGS | E-Deblur-3DGS |

Figure 6: Representative visualization examples of motion deblurring. Note that, our Event-3DGS can be extended for high-quality hybrid reconstruction using events and frames with motion blur.

**Event-3DGS for Color Reconstruction.** In general, adding color information to 3D reconstructed images is crucial for visual appeal and downstream applications. To achieve this, we extend Event-3DGS from a single channel to three channels to enable color reconstruction. As illustrated in Fig 5, we selected a video sequence [15] and utilized the VOLT simulator [22] to convert the RGB channels into events. Using our method framework, we jointly reconstructed these three channels. The results in Fig. 5 demonstrate that our method can achieve high-quality color reconstruction.



| Event | E2VID+3DGS | C-Event3DGS |

Figure 5: Representative examples of colorful event-based 3D reconstruction.

**Limitation.** our method rendering module's adaptive threshold learns a threshold for each scene but doesn't account for variations within the same scene over time. Additionally, our current 3DGS lacks support for dynamic scenarios, where 4DGS may be a solution. Future research will address these limitations to enhance Event-3DGS practicality.

## 5 Conclusion

This paper introduces Event-3DGS, a pioneering event-based 3D reconstruction framework that utilizes 3D Gaussian Splatting (3DGS) to directly process event streams for synthesizing novel views. We present a high-pass filter-based photovoltage estimation module to effectively reduce noise in event data, enhancing the robustness of our method in real-world scenarios. Additionally, we design an event-based 3D reconstruction loss to optimize the parameters of our method. Our results demonstrate that our method surpasses state-of-the-art methods in both reconstruction quality and computational speed on simulated and real-world datasets. We also verify that our method can perform robust 3D reconstruction even in real-world cases with extreme noise, fast motion, and low-light conditions. We believe that our method establishes a new benchmark for using 3DGS with event data, paving the way for high-quality, efficient, and robust 3D reconstruction in challenging real-world scenarios.

Table 5: The influence of the Parameter $\beta$.

| $\beta$ | 0 | 0.1 | 0.15 | 0.2 | 0.25 | 0.5 | 0.75 | 1 |
|---|---|---|---|---|---|---|---|---|
| SSIM | 0.405 | 0.418 | 0.416 | 0.459 | 0.459 | **0.497** | 0.430 | 0.477 |
| PSNR | 8.523 | 8.874 | 8.904 | 10.754 | 10.688 | **12.448** | 9.508 | 11.441 |
| LPIPS | 0.324 | 0.310 | 0.310 | 0.285 | 0.280 | **0.261** | 0.296 | 0.271 |

## Acknowledgements

This work was supported by National Natural Science Foundation of China under Grant 61827804, 62131011.

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

# A   Appendix / Efficiency Advantages over NeRF-based Method

To demonstrate the efficiency of our approach over existing NeRF-based event-based 3D reconstruction methods, we focus on three key aspects: the speed of rasterization compared to ray tracing, the superior parallel support for rasterization in current GPU hardware, and the efficiency of manually derived gradients over neural network backpropagation.

Firstly, the core algorithm of Event-3DGS relies on rendering the 3D Gaussian point cloud using rasterization. Rasterization is inherently faster than ray tracing, which is the method used in NeRF. This speed advantage is crucial for achieving efficient 3D reconstruction.

Secondly, modern GPU hardware is optimized for parallel processing in rasterization. This means that rasterization can be executed with greater efficiency and speed compared to ray tracing, which requires more computational resources and time to process each ray individually. The ability to leverage the parallel processing capabilities of GPUs allows Event-3DGS to perform more efficiently.

Lastly, Event-3DGS utilizes manually derived gradients for optimization, bypassing the complex and time-consuming process of neural network backpropagation used in NeRF. This not only reduces computational overhead but also speeds up the overall process, enabling faster and more efficient 3D reconstruction.

In summary, due to the faster nature of rasterization, better parallel support on GPUs, and the efficiency of manual gradient derivation, Event-3DGS is significantly more efficient than existing NeRF-based methods. This allows for high-quality reconstruction results to be achieved in less time.

# B   Appendix / The Principle of Event-Based Deblurring based on Event-3DGS

To achieve Event-Based Deblurring using Event-3DGS (E-Deblur-3DGS), the key is to associate blurred frames with event data. For a blurry frame camera, its imaging principle can be described as:

$$Y(p, t, C) = \int_{t-t_e}^{t+t_e} \int \lambda \cdot QE_C(\lambda) \cdot L(p, t_i, \lambda) \cdot d\lambda dt_i, \tag{17}$$

where $C$ represents an additional color channel, typically red, green, and blue. $QE_C$ denotes the response magnitude of this channel to different wavelengths of light, while $t_e$ stands for the camera's exposure time, which is the primary cause of motion blur.

Comparing Equation 17 with Equation 4, we see that RGB cameras, while prone to blurring due to their exposure time, offer rich color information. Conversely, event cameras, with their extremely fast response times, capture almost no motion blur but only single-channel spectral information. By combining these strengths through a loss function, motion deblurring can be achieved.

For the event data, we directly apply the loss described in the main text. For the blurry camera, neglecting unit influence, we can sample the output of 3DGS, perform numerical integration, and compare it with the blurry camera's result, expressed as:

$$l_{blur}^{t_1} = \sum_p \frac{|Y(p, t_1, \cdot) - (\int_{t_1-t_e}^{t_1+t_e} G(T(t))dt)|}{W * H}, \tag{18}$$

where $W$ and $H$ are the image's width and height, $T(t)$ is the camera pose at time $t$, and $G(T(t))$ is the 3DGS render result. If the exposure time is not too long and the motion speed is not too fast, the integral can be approximated using numerical integration with a single sample.

Finally, combining the above formulas, the resulting loss is:

$$l_{E-Deblur}(t_1, t_2) = (1 - \alpha_{blur}) \cdot l_{all}(t_1, t_2) + \alpha_{blur} \cdot l_{blur}, \tag{19}$$

where $\alpha_{blur}$ is a hyperparameter controlling the weight of $l_{blur}$. By calculating the gradient with this loss and jointly optimizing the scene and sensor parameters, a clear color image can be successfully reconstructed.

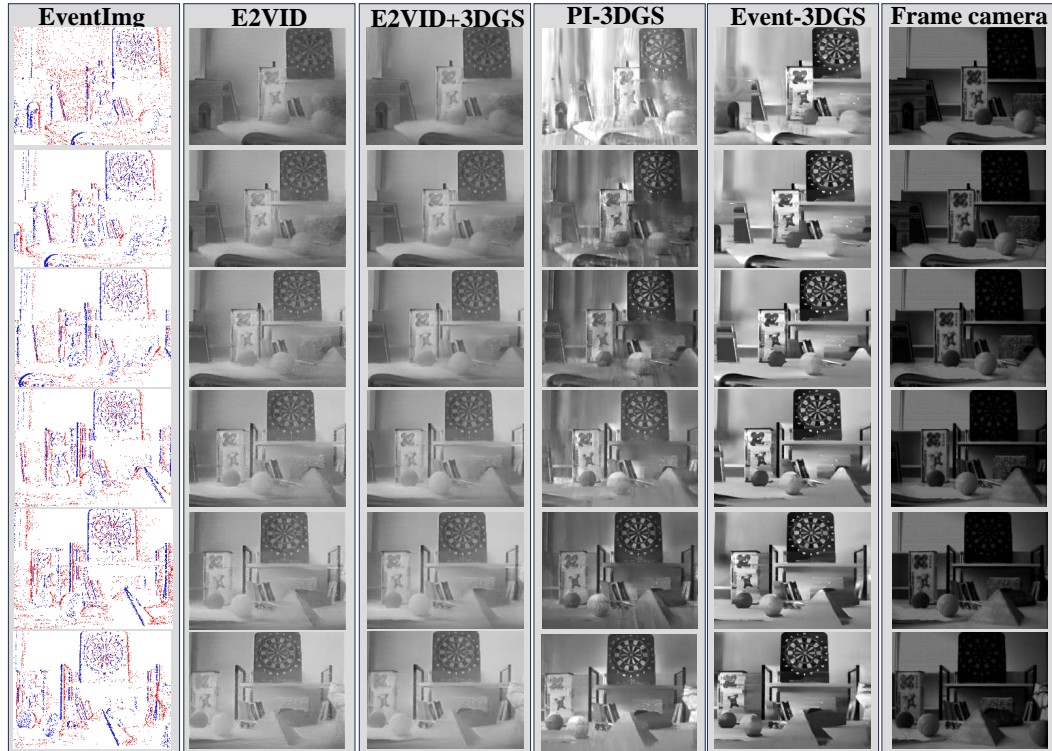

Figure 7: Representative visualization results series the real-world Event-Camera dataset[29]. Our Event-3DGS method produces clearer results compared to other methods.

## C    Appendix / The Principle of Event-3DGS for Color Reconstruction

Since conventional DVS cameras lack color information capture capability, a new sensor type capable of acquiring three-channel events is imperative. Its operational principle is defined as:

$$I_c(p, t, C) = \int \lambda \cdot QE_C(\lambda) \cdot L(p, t, \lambda) \cdot d\lambda, \tag{20}$$

where $C$ signifies an additional color channel, typically denoted as red, green, and blue. $QE_C$ denotes the magnitude of this channel's response to various light frequencies.

Expanding our method algorithm to C-Event-3DGS merely requires extending the loss function. The approach involves computing the loss for each channel's event using the Event-3DGS method and then aggregating these losses to derive the final loss:

$$l_{C-all}(t_1, t_2) = \alpha_R \cdot l_{all}(t_1, t_2)^R + \alpha_G \cdot l_{all}(t_1, t_2)^G + \alpha_B \cdot l_{all}(t_1, t_2)^B, \tag{21}$$

where $l_{all}(t_1, t_2)^R$, $l_{all}(t_1, t_2)^G$, and $l_{all}(t_1, t_2)^B$ represent the loss calculated for each channel's event data using the original Event-3DGS algorithm. The parameters $\alpha_R$, $\alpha_G$, and $\alpha_B$ are hyper-parameters adjusting the weights of each channel. With this loss formulation, C-Event-3DGS can reconstruct a color scene solely based on three-channel event data input. Note that in actual color event cameras, the Bayer pattern is often used to achieve the color effect. Therefore, additional post-processing is required during use, specifically interpolation to obtain the three-channel intensity differences

# D   Appendix / Additional Experiment Result

**Visualization Results on Real Data**. To further showcase the reconstruction prowess of Event-3DGS, we present the reconstruction outcomes from various camera poses within the same scene, denoted as "slider_depth". In Fig. 7, the results illustrate the real Event data reconstruction capabilities of Event-3DGS. Each row in the figure corresponds to the camera positioned in the same pose. As we progress from top to bottom, the camera gradually moves from left to right. Remarkably, Event-3DGS consistently upholds high-quality image reconstruction across a spectrum of poses, underscoring its robust 3D reconstruction capabilities.

**Comparison with Ev-NeRF[14].** To highlight the accuracy superiority of our method over existing approaches, we conducted a comparative analysis with the Ev-NeRF method using a real dataset. Table 6 presents the performance evaluation of our method and Ev-NeRF in three typical real-world scenarios. Observing the results, it's evident that our method consistently outperforms Ev-NeRF across all metrics in the real dataset. This substantial improvement across various scenarios strongly attests to the superiority of our approach.

Table 6: Experiment results on real dataset between Ev-NeRF and our Event-3DGS. Our Event-3DGS outperforms Ev-NeRF in all metrics.

|  | Ev-NeRF | | | Ours Event-3DGS | | |
| --- | --- | --- | --- | --- | --- | --- |
|  | SSIM | PSNR | LPIPS | SSIM | PSNR | LPIPS |
| office_zigzag | 0.415 | 14.559 | 0.275 | 0.430 | 14.043 | 0.183 |
| boxes | 0.470 | 13.979 | 0.320 | 0.575 | 17.696 | 0.259 |
| Dynamic_6dof | 0.260 | 7.100 | 0.420 | 0.212 | 11.404 | 0.307 |
| Average | 0.382 | 11.879 | 0.338 | **0.406** | **14.381** | **0.250** |

**Impact of Training Iterations on Performance and Time**. To empirically demonstrate the time complexity of our proposed Event-3DGS, we investigate the impact of training iterations on its performance and time requirements. Table 7 presents the performance and time metrics of Event-3DGS using synthetic data. The "initial" stage in the table indicates the phase when the parameter $\alpha$ is set to zero. Following this stage, $\alpha$ is adjusted to 0.99, and training proceeds consistently until 7999 iterations. The table illustrates that Event-3DGS achieves high-quality reconstruction results in approximately five minutes. This indicates its potential for real-time applications.

Table 7: Experiment results on performance and time of different training iterations

| iterations | initial | 999 | 1999 | 2999 | 3999 | 4999 | 5999 | 6999 | 7999 |
| --- | --- | --- | --- | --- | --- | --- | --- | --- | --- |
| SSIM | 0.9493 | 0.9528 | 0.9544 | **0.9549** | 0.9546 | 0.9545 | 0.9539 | 0.9530 | 0.9534 |
| PSNR | 23.8661 | 26.5129 | 27.0038 | 27.1565 | 27.2864 | 27.2990 | **27.3594** | 27.1334 | 27.0929 |
| LPIPS | 0.0500 | 0.0448 | 0.0422 | **0.0414** | 0.0428 | 0.0450 | 0.0486 | 0.0509 | 0.0500 |
| Running time(s) | 123 | 141 | 220 | 267 | 314 | 361 | 407 | 453 | 500 |

