# OpenReview forum: "Event-3DGS: Event-based 3D Reconstruction Using 3D Gaussian Splatting"
_NeurIPS.cc/2024/Conference — NeurIPS 2024 poster_

### Official Review · Reviewer_bZpj · 2024-07-01

**Soundness:** 3
**Presentation:** 3
**Contribution:** 2
**Rating:** 6
**Confidence:** 5

**Summary:**

The paper introduces Event-3DGS, the first framework for event-based 3D reconstruction using 3D Gaussian Splatting (3DGS). The method demonstrates superior reconstruction quality, robustness, and efficiency

**Strengths:**

1. By addressing the challenges of fast-motion and low-light scenarios with event cameras and 3DGS, the paper opens new avenues for high-quality, efficient, and robust 3D reconstruction.
2. The quality of the work is demonstrated through extensive experimental evaluations. The authors provide a thorough comparison with state-of-the-art methods, showing significant improvements in reconstruction quality on both simulated and real-world datasets
3. Experiments are conducted thoroughly, including experiments on synthetic and real-world datasets. The ablation study is also conducted

**Weaknesses:**

1. In my opinion, combining 3DGS with Event cameras is somewhat meaningless. The reason is that 3DGS can achieve very high FPS rendering speed and quality, which is essential for real-time tasks like VR and avatar rendering. However, the Event camera modality loses much information, such as color and resolution, which is incompatible with the downstream tasks of 3DGS. In other words, no one would want to experience low-resolution black-and-white VR scenes through a Vision Pro headset.
2. The paper's experimental setup is insufficient, as it does not sufficiently compare reconstruction methods based on Event NeRF. The authors should provide a more comprehensive comparison in Table 2, including [1,2,3]. Furthermore, the datasets used for evaluation are very limited. The authors should include a wider range of real-world data, such as real data capture used in EventNeRF[1].
3. Setting the threshold as a learnable variable and optimizing it together with the 3DGS scene representation may be unreasonable. The threshold dynamically changes with camera movement, scene lighting, and shooting position. Therefore, using a global consistent threshold for all shooting positions is clearly not reasonable. I believe encoding the threshold into (𝑥,𝑦,𝑧,𝜃) for optimization would be more appropriate.

[1] EventNeRF: Neural Radiance Fields from a Single Colour Event Camera

[2] E-NeRF: Neural Radiance Fields from a Moving Event Camera

[3] Ev-NeRF: Event-Based Neural Radiance Field

**Questions:**

1. Can the author provide rendering speed and training times on various datasets? Based on my experience, optimizing 3DGS scene representations often requires more time. Please clarify this.
2. How is the threshold initialized? How can it be optimized with the scene? Is there any difference in the learning rate schedule?
3. In Figure 6, how does the experiment generate blurred images through integration? Can the author more detailed illustration? Besides, I believe that a real-scene blur dataset should be used to support the superiority of Event-3DGS.

**Limitations:**

Combining 3DGS with Event cameras may be ineffective due to the loss of crucial information in Event cameras, such as color and resolution, which are incompatible with the high-quality rendering required for tasks like VR.
Setting the threshold as a global learnable variable for all shooting positions is unreasonable due to dynamic changes with camera movement and lighting; encoding the threshold into (x,y,z,θ) for optimization would be more appropriate.

---

> ### Author Rebuttal · Authors · 2024-08-06
>
> Thank you for your suggestions and affirmation. The figures mentioned below can be found in the attached PDF.
>
> W1: Combining 3DGS with Event cameras is somewhat meaningless.
>
> R1: The combination of these two approaches is meaningful:
>
> (1) Pure event data excels in SLAM, 3D reconstruction, and autonomous driving, especially in high-speed or low-light scenes where it outperforms RGB frames.
>
> (2) For high visual quality, our method can be extended to a multimodal framework combining RGB and event data. This joint approach is useful for tasks like high-speed motion de-blurring (see Fig. 6) and HDR imaging.
>
> W2: A more comprehensive comparison, including EventNeRF, E-NeRF, and Ev-NeRF.
>
> R2: Per your suggestion, Ev-NeRF and EventNeRF can be used for comparison, but E-NeRF has a bug that prevents it from being tested.
>
> (1) For Ev-NeRF, we report experimental results on both synthetic datasets (see Table R1 and Fig. R1) and real-world datasets (see Table R2  and Fig. R2). Note that, our Event-3DGS significantly outperforms Ev-NeRF in terms of SSIM, PSNR, and LPIPS.
>
> Table R1: Comparison of Ev-NeRF and our Event-3DGS in the synthetic dataset
>
> |             ||Ev-NeRF||           |ours||
> |-------------|------|-------|------|------|-----------|------|
> |             |SSIM|PSNR|LPIPS|SSIM|PSNR|LPIPS|
> |**mic**|0.858|16.939|0.316|0.952|21.127|0.063|
> |**ship**|0.656|16.521|0.398|0.818|17.815|0.147|
> |**materials**|0.692|11.272|0.482|0.933|20.506|0.060|
> |**lego**|0.743|18.526|0.303|0.925|23.046|0.058|
> |**ficus**|0.794|13.368|0.244|0.940|19.939|0.049|
> |**drums**|0.797|16.793|0.315|0.951|22.568|0.042|
> |**chair**|0.786|10.945|0.239|0.953|27.336|0.050|
> |**average**|0.761|14.909|0.328|**0.925**|**21.762**|**0.067**|
>
>  Table R2: Comparison of Ev-NeRF and our Event-3DGS in the real-world dataset.
>
> |                      ||**Ev-NeRF**||           |**Ours**||
> |----------------------|------|----------|------|------|-----------|------|
> |                      |SSIM|PSNR|LPIPS|SSIM|PSNR|LPIPS|
> |**slider_depth**|0.282|6.033|0.442|0.497|12.448|0.261|
> |**outdoors_walking**|0.208|8.595|0.409|0.271|10.583|0.300|
> |**calibration**|0.114|12.023|0.712|0.312|11.065|0.222|
> |**average**|0.201|8.883|0.521|**0.360**|**11.365**|**0.261**|
>
> (2) For EventNeRF, it processes color event data unlike our format. Thus, we converted its synthetic data to our classic event data format. Table R3 and and Fig. R3 shows that EventNeRF and our Event-3DGS perform comparably. Note that, EventNeRF’s data differs from the original paper due to additional color adjustments made before metric calculation. This adjustment may be not entirely reasonable, so we compare only the raw reconstructed images in the two methods. Besides, while EventNeRF requires a long training time for each scene, our Event-3DGS trains each sequence in under 10 minutes. In short, Event-3DGS offers better efficiency while maintaining comparable reconstruction quality.
>
> Table R3: Comparison of EventNerf and our Event-3DGS in the synthetic dataset.
> |           ||EventNeRF||           |**Ours**||
> |-----------|------|----------|------|------|--------|------|
> |           |SSIM|PSNR|LPIPS|SSIM|PSNR|LPIPS|
> |**lego**|0.894|23.137|0.076|0.914|23.694|0.111|
> |**drums**|0.923|26.917|0.054|0.942|24.394|0.055|
> |**chair**|0.952|28.905|0.047|0.958|24.449|0.035|
> |**average**|0.923|**26.320**|**0.059**|**0.938**|24.179|0.067|
>
> (3) For E-NeRF, we attempt deployment but faced insurmountable issues. A review of related GitHub discussions reveal that others have encountered similar problems without resolution from the authors, leading us to abandon this comparison.
>
> Overall, we have tested many sequences with high-noise real event data, proving its practicality. The real object data used in EventNeRF comes from a different color event camera, which makes it unsuitable for our algorithm.
>
>
> W3: Making the threshold a learnable variable and optimizing it with the 3DGS scene representation may be unreasonable.
>
> R3: Emphasizing adaptive thresholds is valuable. We explore a basic approach using an adaptive threshold for a single sequence. While varying scene or pixel-level thresholds could improve reconstruction quality, this would likely increase computational complexity, which we acknowledge as a limitation. These points are important and could be explored further in future work.
>
> Q1: Can the author provide rendering speed and training times on various datasets?
>
> A1: Our Event-3DGS optimization is fast. For instance, using the Ficus sequence, we achieved the reported results in approximately 9 minutes and 47 seconds on an RTX 3080 Ti. However, the original 3DGS has high GPU memory requirements, and running out of memory can significantly slow down subsequent reconstructions.
>
> Q2: How is the threshold initialized? How can it be optimized with the scene?
>
> A2: We can start by setting the threshold to the manufacturer’s standard value or an estimated range (typically between 0 and 1). During training, this threshold can then be optimized as a learnable parameter alongside the scene parameters. This approach differs from a learning rate schedule, which is a manually designed strategy not directly tied to backpropagation or training but helps facilitate the training process. We will add these details in the revised version.
>
> Q3: How does the experiment generate blurred images through integration in Fig. 6?
>
> A3: To accurately simulate the blur effect, we start with a high-quality 3D model and render the blurred images. During rendering, we calculate the spectral integration of the camera's motion based on its speed to produce the final blurred image. Given the ground truth data of the 3D scene distribution, the resulting blurred images closely approximate real-world conditions.

---

> > ### Comment · Reviewer_bZpj · 2024-08-11
> >
> > The author's response has nicely addressed my concerns, and I have increased my rating. Good luck!

---

> ### Author Response · Authors · 2024-08-11
>
> Thank you for your recognition of our work and for raising your score from a 5 to a 6. We appreciate your constructive feedback and support, which have been instrumental in improving our research.

---

### Official Review · Reviewer_nxFk · 2024-07-12

**Soundness:** 2
**Presentation:** 1
**Contribution:** 2
**Rating:** 5
**Confidence:** 5

**Summary:**

The paper introduces Event-3DGS, a framework that leverages event cameras and 3D Gaussian Splatting (3DGS) for efficient and robust 3D reconstruction in challenging real-world scenarios. The authors propose: a high-pass filter-based photovoltage estimation module and a novel event-based 3D reconstruction loss to enhance performance. Experiments with both real and synthetic datasets are used to evaluate the performance of the proposed method. The results demonstrate it performs better than prior works.

**Strengths:**

1: The combination of event camera with 3D-GS is interesting and enable the leverage of efficient rendering capability of 3D-GS;

2: The experimental results, which conduct on both real and synthetic datasets, demonstrate the proposed method performs better than baseline methods, in terms of visual quality;

3: Additional experiments which integrate the proposed method with motion blurred RGB images, as well as for color reconstruction, are presented; The experiments demonstrate its advantage for more practical applications.

**Weaknesses:**

1: The paper is not well written and contains many errors, it is very difficult to follow for certain parts. For example:
* why it holds for Eq. 9, based on the reviewer's understanding, the uncertainty term or noise term is usually a random gaussian noise. why it equals to the negative of E?
* the notation is confusing, V is used to denote signal after Laplace transformation, while it is used as photovoltage and photocurrent contrast in Eq. 10.
* Eq. 12 conflicts with Eq. 10, why the same term, i.e. \hat{V}_d, can equal to E's integration in Eq. 10 and E + plus a differential term in Eq. 12?
* It is confusing on "when the time is sufficiently close, we consider the relationship between time t and pose T as a one to one mapping" in Line 127.
* Line 147, "V_th is a fixed constant" -> typos, same for Eq. 7.

2: Experimental evaluations against prior methods are not sufficient. The paper exploits E2VID, E2VID+3DGS, PI-3DGS as the baselines. However, it misses several important baselines which are on event based NeRF. Although the authors present additional comparisons against EvNeRF in the appendix. It misses the comparisons on the synthetic datasets. The evaluations on real data against EvNeRF also misses some sequences which presented in Table 2.

3: In terms of the ablation studies on the contribution of each component, why the authors chose the baseline that integrates E2VID and 3DGS for event based 3D reconstruction. It should be experiments for the proposed method instead of E2VID.

4: Dependency on known poses: the approach requires ground truth poses, which limit its practical usage of the proposed method. Different from frame-based methods, which can exploit COLMAP to obtain GT poses, it is usually difficult to obtain GT poses for event camera alone.

Main reasons on current rating:

Based on the poor presentation of the paper, and insufficient evaluations to fully demonstrate the advantages of the proposed method, the reviewer gives his current rating.

**Questions:**

n.a.

**Limitations:**

n.a.

---

> ### Author Rebuttal · Authors · 2024-08-06
>
> The figures (i.e., Fig. R1 and Fig. R2) mentioned below can be found in the attached PDF.
>
> Q1: The paper is not well written and contains many errors, it is very difficult to follow.
>
> A1: Thanks. We will clarify the writing to improve reader understanding in the camera-ready version.
>
> Q1-1: In Eq. 9, why it equals to the negative of $E$?
>
> A1-1: Eq. 9 is a parameter set used to simplify assumptions, which is convenient for the theoretical derivation and representation of subsequent filtering. $w$  should not be treated as a Gaussian noise term. The negative term is reasonable, as it accommodates step-like changes in light intensity, which are typically within the feasible range of  $w$ . We will clarify this explanation in the revised version.
>
> Q1-2: $V$ is used to denote signal after Laplace transformation.
>
> A1-2: The notation $V$ should indeed be different after the Laplace transform. We will correct this in the camera-ready version.
>
> Q1-3: Eq. 12 conflicts with Eq. 10.
>
> A1-3: Eq. 10 represents the pure integration method, which estimates light intensity differences under ideal conditions but performs poorly in noisy environments. In contrast, Eq. 12 employs high-pass filtering, making it more robust to noise. Thus, Eq. 10 and Eq. 12 are two different approaches for estimating light intensity differences and are not contradictory. We will add the details in the revised version.
>
> Q1-4: It is confusing on "when the time is ..." in Line 127.
>
> A1-4: The camera pose $T$ can be regarded as a function of time $t$ , which is generally not one-to-one. For example, a camera might return to the same position with the same orientation at different times. However, within a sufficiently small time interval, this function can be treated as one-to-one, meaning each pose corresponds to a unique point in time. We will provide a more detailed description of this in Line 127 in the revised version.
>
> Q1-5: Line 147, the typo minor.
>
> A1-5: The omission of the subscript was a mistake, and we will correct it in the revised version.
>
> Q2: Experimental evaluations against prior methods are not sufficient. It misses the comparisons on the synthetic datasets. The evaluations on real data against Ev-NeRF also misses some sequences in Table 2.
>
> A2: Thanks for your comments. We have conducted additional experiments comparing our Event-3DGS with Ev-NeRF on both synthetic (see Table R1) and real-world datasets (see Table R2). Our Event-3DGS significantly outperforms Ev-NeRF in terms of SSIM, PSNR, and LPIPS. Additionally, we provide representative visualization examples for both the synthetic data (see Fig. R1) and the real-world dataset (see Fig. R2). Note that, we have added 3 additional sequences in the real dataset that are not shown in Table 6 of the appendix. These results demonstrate that Event-3DGS achieves superior reconstruction quality compared to Ev-NeRF. We will present these experimental results for Ev-NeRF in the camera-ready version.
>
> Table R1: Comparison of Ev-NeRF and our Event-3DGS in the synthetic dataset.
>
> |             ||Ev-NeRF||           |ours||
> |-------------|------|-------|------|------|-----------|------|
> |             |SSIM|PSNR|LPIPS|SSIM|PSNR|LPIPS|
> |**mic**|0.858|16.939|0.316|0.952|21.127|0.063|
> |**ship**|0.656|16.521|0.398|0.818|17.815|0.147|
> |**materials**|0.692|11.272|0.482|0.933|20.506|0.060|
> |**lego**|0.743|18.526|0.303|0.925|23.046|0.058|
> |**ficus**|0.794|13.368|0.244|0.940|19.939|0.049|
> |**drums**|0.797|16.793|0.315|0.951|22.568|0.042|
> |**chair**|0.786|10.945|0.239|0.953|27.336|0.050|
> |**average**|0.761|14.909|0.328|**0.925**|**21.762**|**0.067**|
>
>
> Table R2: Comparison of Ev-NeRF and our Event-3DGS in the real-world dataset.
>
> |                      ||**Ev-NeRF**||           |**Ours**||
> |----------------------|------|----------|------|------|-----------|------|
> |                      |SSIM|PSNR|LPIPS|SSIM|PSNR|LPIPS|
> |**slider_depth**|0.282|6.033|0.442|0.497|12.448|0.261|
> |**outdoors_walking**|0.208|8.595|0.409|0.271|10.583|0.300|
> |**calibration**|0.114|12.023|0.712|0.312|11.065|0.222|
> |**average**|0.201|8.883|0.521|**0.360**|**11.365**|**0.261**|
>
> Q3: In terms of the ablation studies on the contribution of each component, why the authors chose the baseline that integrates E2VID and 3DGS for event based 3D reconstruction. It should be experiments for the proposed method instead of E2VID.
>
> A3: Sorry for the writing mistake that may have misled you. We confirm that the data in Table 3 is correct, but the description of the baseline in Lines 250-251 is incorrect. The baseline should be described as a pure integration image without the adaptive threshold and event loss, not E2VID+3DGS. Additionally, comparing Table 2 and Table 3 shows that the ablation baseline performance in Table 3 does not match the E2VID performance in Table 2, confirming that the baseline is not E2VID. We will correct the baseline description in Lines 250-251 in the camera-ready version.
>
>
>
> Q4: Dependency on known poses. It is usually difficult to obtain GT poses for event camera alone.
>
> A4: In general, obtaining accurate camera poses in normal scenes using only event cameras is challenging compared to RGB frames, due to the sparse sampling of event cameras. However, recent works [1, 2] show that event cameras can be effective for pose estimation, particularly in challenging scenarios such as high speeds or low light. For example, Muglikar et al. [1] developed the E2Calib tool, which uses E2VID to reconstruct gray frames, calibrate camera extrinsics, and estimate poses. In our work, we use E2VID to reconstruct frames and then apply COLMAP to process frames to estimate camera poses. Thus, the camera poses estimated using event data enable our Event-3DGS to produce high-quality reconstructions in real-world applications.
>
> [1] How to calibrate you event camera, CVPRW 2021.
>
> [2] EVO: A geometric approach to evnet-based 6-DOF parallel tracking and mapping in real time, IEEE RAL 2017.

---

> ### Comment · Reviewer_nxFk · 2024-08-13
>
> Thanks for the effort in detailed rebuttal. My questions are addressed and I would like to upgrade my rating. It is suggested to further improve the writing and incorporate all the new experimental results into the final version.

---

> > ### Author Response · Authors · 2024-08-13
> >
> > Thank you very much for your thoughtful feedback. We deeply appreciate your time and detailed comments. We will carefully address your suggestions by enhancing the writing and integrating the new experimental results into the final version. Your support is invaluable to us.

---

### Official Review · Reviewer_tU43 · 2024-07-12

**Soundness:** 3
**Presentation:** 3
**Contribution:** 3
**Rating:** 7
**Confidence:** 3

**Summary:**

This paper proposes a 3D reconstruction method from event cameras using 3D Gaussian Splatting (3DGS). The authors propose an innovative framework that takes a stream of events as input and optimize 3D appearance model with 3DGS. In particular, the authors propose differentiable rendering of event images and photovoltage contrast image with corresponding losses. Using photovoltage contrast images allow to handle noisy inputs with make the proposed method applicable to real data.

**Strengths:**

-	The paper is well written and easy to follow. Although I am not expert in the domain, I found the proposed method interesting and novel (as far as I know). The high pass filtering technique to handle noisy images makes much sense and it is also modeled in the rendering of Gaussians Splatting. The proposed losses seem correct and allow to achieve state-of-the-art results.

-	The experimental section is convincing. The results are clearly superior to previous methods and the ablation part shows that the losses are working well. The authors also propose a concrete application for the task of image deblurring.

**Weaknesses:**

-	In the writing, it is better to avoid abbreviations like “it’s” and write “it is” (l. 129 for example).

-	A word is missing l. 147

-	In equation (7), I cannot understand the meaning of i^p. I understand from l. 139 that p is 2D pixel coordinate, but I am not sure what is an integer at the power of a 2-dimensional vector.

-	In the experiments it seems that numbers in the text l.230 do not match the numbers in table 2.

**Questions:**

Please clarify equation (7)

**Limitations:**

Yes limitations such as dynamic scenes have are included

---

> ### Author Rebuttal · Authors · 2024-08-06
>
> Thanks for your positive evaluation.
>
> W1: In the writing, it is better to avoid abbreviations like “it’s” and write “it is”.
>
> R1: we will add the missing words and change “it’s” to “it is” in the camera-ready version.
>
> W2: A word is missing in Line 147.
>
> R2: We missed "event sensor" in Line 147, and will correct it in the camera-ready version.
>
> W3: Questions about Equation (7).
>
> R3: Our intention is to use $t$ as the subscript and $p$ as the superscript to represent the time when the $i$-th event is triggered at pixel $p$, without implying an exponent. We will correct this in the camera-ready version.
>
> W4: It seems that numbers in the text Line 230 do not match the numbers in Table 2.
>
> R4: Actually, Line 230 corresponds to the numbers in Table 1, not Table 2. The statement "our Event-3DGS has improved by 0.017 and 2.227, respectively" refers to the performance metrics for the synthetic dataset shown in Table 1, not the real-world dataset in Table 2.

---

> > ### Comment · Reviewer_tU43 · 2024-08-12
> >
> > I have read the authors' rebuttal and the other reviews. The authors have answered my questions and I am satisfied with the response. This is an interesting paper. I keep my initial rating of accept.

---

> > > ### Author Response · Authors · 2024-08-12
> > >
> > > Thank you sincerely for recognizing the value of our work and for your positive feedback. We are especially grateful that you found our research both interesting and worthy of consideration. Your recommendation for acceptance is a meaningful affirmation, and we truly appreciate the time and care you invested in evaluating our submission.

---

### Official Review · Reviewer_kCE4 · 2024-07-20

**Soundness:** 2
**Presentation:** 2
**Contribution:** 3
**Rating:** 5
**Confidence:** 4

**Summary:**

This paper proposes a new method for novel view synthesis of intensity images using Event-Camera data via 3d Gaussian Splatting. Event-cameras are a type of camera that captures log intensity changes on the image plane. While 3D Gaussian splatting is a method for novel view synthesis that has been originally developed for images from traditional cameras. Thus, the overall achievement of this work is to make 3D Gaussian splatting work for asynchronous event data.

To achieve this, the authors introduce a high-pass filter for noise reduction in the photovoltage contrast estimation, a technique for integrating event data in the rasterization process, and a two component loss term including a proposed schedule for the loss terms parameters. The work is evaluated on a synthetic and a real-world dataset and compared against an event-based video generation and its integration with the image-based Gaussian splatting.

**Strengths:**

Integrating event-camera data with Gaussian Splatting is an interesting research direction. The work makes a meaningful technical contribution that are of interest for both, the event-vision and novel view synthesis sub-communities. I also want to applaud the authors for committing to releasing the source code making the reproduction of this work easy. From the work itself, I liked the idea of introducing a filter and the loss design to the extent that I understood it.

**Weaknesses:**

The work is currently written in a way that is more amenable to the vision community and might need some rewriting to be understandable by the broader NeurIPS audience, which e.g. cannot be assumed to be familiar with Gaussian Splatting. Even if space does not permit to add a full background section (which would be desirable), it would be useful to at least introduce $G(\mathbf{T})$ in a way that gives a high-level undstanding of what Gaussian Splatting does. The current writing merely says "For a specific 3D scene represented by 3D gaussian points, the forward process of 3DGS can be regarded as a mapping function G(T)" which is probably not enough for a reader unfamiliar with 3DGS. There are more examples like these where average NeurIPS readers might need more background or better explanations than those at vision venues.

Some terminology in the paper is used wrongly or made up which makes it hard to understand. E.g. the authors speak of 3D Gaussian points which is not a common term and in this context wrong as 3DGS uses 3d Gaussian functions (or simply 3D Gaussians) rather than points. A more nuanced point is that 3DGS is referred to as a reconstruction method. And although its explicit representation is much closer linked to reconstruction than the neural network-based radiance fields, I would carefully argue that Novel View Synthesis (NVS) would be the more correct term. The evaluation of the work also does not focus on reconstruction but on NVS.

The work seems not to compare against any Event-based NeRF techniques which one would expect due to the conceptual similarity. Moreover I wonder whether the proposed methodology could also be used for some of the recent NeRF techniques. Unless, I overlooked something, it seems like the work does not make any use of the fact that the underlying representation is Gaussian. So, it would be interesting to see how much of the performance is due to the authors ideas and how much is due to the use of 3DGS as underlying representation.

Minor points:
* The authors often speak of "our Event-3DGS" in reference to their method. I am not sure this is grammatically correct. Shouldn't this be rather "our method" or "our approach"?
* The research gap mentioned in l.25/26 is not a fully open reseach problem in the generality stated there. It's rather that specific aspects of it are unsolved.
* l.34/35 "Traditional non-learning optimized-based methods" -> "Traditional non-learning optimization-based methods"
* l. 39-42 slightly too long sentence should maybe be split. Also "achieves" -> "achieve"
* The structure of the contribution list is confusing. Since the first bullet point encompasses the other two. I would suggest to focus on the technical contributions and maybe leave out the first point. The content of the first point could still be acommodated in the previous paragraph.
* l.69 / 70 "often struggle to achieve robustness and high-quality reconstruction" -> "often struggle to achieve robust and high reconstruction quality" maybe?
* l.78/79 I am not sure whether some of the statements about NeRFs are true. Modern NeRFs have shown significant improvement in training time. Isn't the main advantage of splatting the fast view generation?
* In Fig 1., is there a reason why $\frac{1}{\theta}$ is also dark red in the "Photovoltage Contrast Rendering" pane? Also, shouldn't  "Symbol" be replaced by "Symbols" or "Legend"?
* In l.109, should the second "T" be bold? Also, the authors do not spell out how poses are parametrized. I assume it is the same as in 3DGS?
* The work speaks of "forward process of 3DGS"
* The letter $\alpha$ is used in two different contexts (l.110 and eq 16) and, thus, overloaded which may create confusion.
* l. 114/115 "integration of asynchronous events into raw 3DGS" -> "integration of asynchronous events into the original 3DGS formulation"
* In eq. 2, is it work including $p$ instead of writing $\cdot$ as a placeholder for the pixel location?
* Can an explanation of photovoltage be added to the paper? At least, maybe the explanation of the image formation in event cameras at the beginning of Sec. 3.5 should come earlier in the paper.
* Some subscripts are broken in l. 147 and eq. 7
* In eq 8. does $w(p,t)$ also encode the quantization error?
* l. 162 "we can simply assume as" -> "we can simply assume"
* Also the typesetting of $\log$ is not consistent. It is italic in eq. 3 and upright in l. 127. I recommend using the latter to e consistent with the $\max$ in l.197
* Is it woth renaming Sec. 4.2. to "Results" and Sec. 4.3 to "Ablation Studies"?
* Some of the advertising (to the extent that it even should be in the paper) in l. 292-296 seems to be more suited for the conclusion.

**Questions:**

* In l. 112, what is meant by "point mapped to this pixel"? The Gaussian function, its center, or a point on its surface?
* In l.200 it is said that "theoretical analysis shows that setting $\beta$ to 0.5 yields excellent results". I am not sure I have seen any theoretical analysis. Did I overlook? If not, could the authors elaborate on this?
* What is the intuition for setting $\alpha$ to 0 in the beginning of the optimization? Does it speed up training time? Which other training schedules have been tried? What happens if it has a non-zero value already from the beginning of the process?
* How were the datasets used? Do the scenes already contain a test validation split or was it created by the authors? What was the length of event-sequences that were sampled during training? I.e. what is the value of $N$ from eq. 7 during training.
* What is the impact of different values for $\tau$? Is there a reason why it is set to 0.05?
* What is meant by "$E(p,t)$ is ideal" in l. 164?
* What is the intuition of high pass filtering. Is the noise assumed to be low frequency?
* What is the purpose of the subscript $i$ in eq. 14. It seems to not appear on the right hand side.
* l. 253: in what sense is the threshholding adaptive?
* Looking at Table 7 in the appendinx, to the authors have an intuition on why the performance goes down as training progresses?
* Sec. 4.4. seems to introduce a new method that combines events and images rather than be a "Scaling test". Could you provide some information on how the combined method works?
* Also, how are the visualizations created? Using purely events does not yield a baseline color and the intensity might be off by a constant offset. How is this resolved in the work?

**Limitations:**

Overall, the work mentions most limitations. One that is currently omitted is that the way it incorporates colors is not realistic in practice. If understood correctly, the work assumes color information to be equal in each pixel. However, in practice color event cameras use a Bayer RGB pattern and require some sort of remosaicing during the splatting process.

---

> ### Author Rebuttal · Authors · 2024-08-07
>
> Thanks for you insightful suggestions and affirmation. The tables and figures mentioned below can be found in the attached PDF.
>
> W1: Rewriting should be clear for the NeurIPS audience.
>
> R1: We will add a high-level introduction to Gaussian Splatting in the main manuscript and provide clearer explanations in the appendix of the revised version.
>
> W2: Some terminology may be hard to understand. Not focus on reconstruction but on NVS.
>
> R2: Thanks. We will clarify the terminology in the revised version.
>
> Besides, existing 3DGS methods mainly evaluate NVS to align with common image reconstruction quality metrics. Therefore, we also use NVS to assess our Event-3DGS. However, our method is not restricted to NVS and can be extended to obtain depth and reconstruct 3D meshes.
>
> W3: The work seems not to compare against any NeRF techniques. Please explain the performance contributions.
>
> R3: First, we compare our Event-3DGS with a typical method (i.e., Ev-NeRF) on both synthetic datasets (see Table R1 and Fig. R1) and real datasets (see Table R2 and Fig. R2). Note that, our Event-3DGS significantly outperforms Ev-NeRF in terms of SSIM, PSNR, and LPIPS.
>
> Second, our Event-3DGS benefits from both 3DGS itself and our proposed modules (see the ablation test in Table 3). Unlike NeRF’s ray tracing, 3DGS uses raster-based rendering to generate the entire image at once, which can introduce errors if many pixels are not sampled when events occur. Our loss design and filtering techniques address these errors, improving performance.
>
> W4: Some writing minor points.
>
> R4: Thanks. We will address these writing minors one by one.
>
> Q1: In Line 112, what is meant by "point mapped to this pixel"?
>
> A1: The transparency of the 3D Gaussian function is first mapped to screen space. Next, pixel transparency is determined based on each pixel's position within the 3D Gaussian function. The color is determined from the spherical harmonics lighting coefficients. We will clarify this description in the revised version.
>
> Q2: In Line 200, elaborate on the theoretical analysis.
>
> A2: An event camera cannot capture intensity information between events, and the intensity can vary within the threshold range (see Eq. 8). The parameter selection in Line 200 considers this, ensuring that the rendered intensity within the positive and negative threshold range minimizes the loss, aligning with the theoretical analysis. We will add this analysis in the camera-ready version.
>
> Q3: The setting and optimization of $\alpha$.
>
> A3: $\alpha$ is a hyperparameter chosen to provide a good initial value for 3DGS. It doesn't need to be exactly 0, and any small value can work. While setting $\alpha$ to the maximum value is possible, it may lead to local optima issues.
>
> Q4: The datasets setting and the length of the event sequences sampled.
>
> A4: We split each synthetic or real dataset, reserving a portion of the images for testing while using the rest for reconstruction training. The event sequences vary in length, but generally contain more than a million events. We will add these details in the experimental settings in the revised version.
>
> Q5: What is the impact of different values for $\tau$? Is there a reason why it is set to 0.05?
>
> A5: This hyperparameter $\tau$ controls the frequency range of the high-pass filter. A narrower range reduces noise but may discard more original event information. Besides, our tuning experiments show that setting this parameter to 0.05 yields better results.
>
> Q6: What is meant by "$E(p, t)$ is ideal" in Line 164?
>
> A6: "The ideal" means that an event perfectly represents intensity changes without any noise. In practice, events are affected by background noise, refractory periods, leak noise, and bandwidth limitations. We will provide more details in the revised version.
>
> Q7: What is the intuition of high pass filtering. Is the noise assumed to be low frequency?
>
> A7: Yes, due to the higher noise levels in event cameras compared to traditional RGB cameras, especially in low-light scenes, the noise primarily consists of low-frequency signals compared to the effective events.
>
> Q8: What is the purpose of the subscript $i$ in Eq. 14.
>
> A8: The symbol used here seems inappropriate, as it was intended only to differentiate it from the subsequent event loss. We will correct this in the camera-ready version.
>
> Q9: Line 253, in what sense is the threshold adaptive?
>
> A9: The threshold for detecting light changes in event cameras may not be directly obtained from event data, and it can be influenced by the scene environment (see Lines 148-150). Accurate threshold estimation is crucial for obtaining light intensity from a mathematical optimization perspective. To improve reconstruction quality, we design an adaptive threshold for each scene (see Table 3).
>
> Q10: Table 7, goes down as training progresses.
>
> A10: The training has essentially converged at a certain point, with subsequent steps involving only minor perturbations. In practice, although there are fluctuations after convergence, the overall performance remains stable.
>
> Q11: Explain how the combined method.
>
> A11: We have provided the combination of events and frames in the appendix (see Lines 469-488). Briefly, we compare the rendered image with the RGB frame to obtain a loss, which is then added to the total loss (see Eq. 16 and Eq. 19).
>
> Q12: Visualizations created.
>
> A12: In pure event-based imaging, restoring absolute light intensity is generally not possible. We specify the initial light intensity (e.g., setting the background intensity) as done in previous methods. Typically, this value can be set to 0, representing pure black.
>
> Q13: Colors is not realistic.
>
> A13: We assume each pixel has three channels of events, differing from the Bayer pattern. Therefore, our algorithm first interpolates the raw data into a three-channel format. In the future, we could extend our Event-3DGS to handle raw color event data in the Bayer pattern.

---

> ### Comment · Reviewer_kCE4 · 2024-08-12
>
> Thank you for your responses.  Many of my points have been addressed and I still see the paper still on the accept side. The reason I tend to not increase my score further is that some of the promised changes require a significant change in writing and it is hard to judge those without having another full round of reviews.

---

> > ### Author Response · Authors · 2024-08-13
> >
> > Thank you for your detailed review and thoughtful suggestions. Your careful analysis has been invaluable in helping us improve our work. We will ensure that the writing concerns you highlighted are addressed in the camera-ready version. Additionally, we will further open the relevant code to enhance transparency and reproducibility. We greatly appreciate the time and effort you dedicated to reviewing our manuscript.

---

### Author Rebuttal · Authors · 2024-08-07

Thank you to the area chairs and all the reviewers for your valuable comments!

To further verify the effectiveness of our Event-3DGS, we conducted six experimental tests using two event-based NeRF methods (i.e., Ev-NeRF [1] and EventNeRF [2]). The quantitative results and visualization figures are available in the attached PDF file. Specifically:

- **Table R1** and **Table R2** provide a quantitative comparison of Ev-NeRF and our Event-3DGS on synthetic and real-world datasets.

- **Fig. R1** and **Fig. R2** showcase representative visualization examples for Ev-NeRF and our Event-3DGS on synthetic and real-world datasets.

- **Table R3** and **Fig. R3** show the quantitative and visual experimental results of EventNeRF and our Event-3DGS, respectively.

For convenience, we highlight the tables and figures relevant to each reviewer's comments as follows:

- **Reviewer kCE4**: Table R1, Table R2, Fig. R1, and Fig. R2.

- **Reviewer nxFk**: Table R1, Table R2, Fig. R1, and Fig. R2.

- **Reviewer bZpj**: Table R1, Table R2, Table R3, Fig. R1, Fig. R2, and Fig. R3.

We have provided point-by-point responses to each reviewer's comments in the corresponding rebuttal. Please review these responses.

Thank you again for your insightful comments. We sincerely hope the rebuttal will address the reviewers' concerns and convince the reviewers to give more convincing decisions.



[1] Ev-NeRF: Event-Based Neural Radiance Field, WACV 2023.

[2] EventNeRF: Neural Radiance Fields from a Single Colour Event Camera, CVPR 2023.

---

### Decision · Program_Chairs · 2024-09-25

**Decision:**

Accept (poster)

**Comment:**

All reviewers agreed to accept the paper. The paper originally got BA, A, BR, and BA, and, after the rebuttal period, the last two reviewers raised the ratings, leading to final ratings of BA, A, BA, and WA.

Reviewers recognized that the paper has clear contributions in applying 3DGS for event cameras, considered the first in the field.
However, reviewers are also concerned about the unsatisfactory readability and issues in evaluations. After the authors' rebuttal and discussion period, most reviewers’  concerns have been resolved, and all reviewers agreed that the paper's shape is above the bar for NeurIPS.

The AC also supports accepting the paper and strongly suggests the authors to follow the reviewers’ feedback in the camera-ready version.